# Tumor Biological Feature and Its Association with Positive Surgical Margins and Apical Margins after Radical Prostatectomy in Non-Metastasis Prostate Cancer

**Shuo Wang †, Peng Du \*, Yudong Cao, Xiao Yang and Yong Yang**

Key Laboratory of Carcinogenesis and Translational Research (Mninistry of Education), Urological Department, Peking University Cancer Hospital & Institute, Beijing 100142, China; wangshuoarea@pku.org.cn (S.W.); ydcao@bju.edu.cn (Y.C.); yangxiao@126.com (X.Y.); yoya_urology@sina.com (Y.Y.)

\* Correspondence: dupeng9000@126.com
† First author: Shuo Wang.

**Abstract:** Purpose: We assessed clinical and tumor biological features and evaluated their association with positive surgical margins (PSMs) and positive apical margins (PAMs) variability after radical prostatectomy (RP) in men with non-metastasis prostate cancer (nmPCa) in our institute. Patients and methods: During the period from January 2013 to December 2017, clinical and pathological data were collected in 200 patients with nmPCa undergoing RP in the Urological department of Peking University Cancer Hospital & Institute. Surgical and apical margins were stated negative and positive, separately. A dichotomous logistic regression model was used to assess clinical and tumor biological features including age, total prostate volume (TPV), biopsy positive cores (BPC), D'Amico risk grade, tumor clinical stage, International Society of Urologic Pathology (ISUP) grade, tPSA, f/t and pelvic lymph nodes (PLN) invasion, and their association with PSMs and PAMs was evaluated. Results: Overall, men with nmPCa in this study had a high ISUP grade (58.5% grade 3–5), high risk grade (89.4%) and high clinical T stage (56% cT3-4). PSMs were detected in 106 patients; the rate of PSMs was 53%. Among patients with PSMs, 83% were PAMs; the overall rate of PAMs was 44%. Among patients with PSMs, high risk (OR, 1.439; $p = 0.023$), cT3a (OR, 1.737; $p = 0.045$), cT3b (OR, 5.286; $p < 0.001$), cT4 (OR, 6.12; $p < 0.001$), ISUP Grade 4 (OR, 2; $p = 0.034$) and Grade 5 (OR, 6.167; $p < 0.001$) and PLN invasion (OR, 6; $p = 0.019$) were strongly associated with PSMs using a dichotomous logistic regression univariable model, and high risk (OR, 6; $p = 0.019$), cT3a (OR, 5.116; $p = 0.048$), cT3b (OR, 9.194; $p = 0.008$), cT4 (OR, 4.58; $p = 0.01$), ISUP Grade 4 (OR, 7.04; $p = 0.035$), Grade 5 (OR, 16.514; $p = 0.002$) and PLN invasion (OR, 5.516; $p = 0.03$) were independently associated with PSMs by using multivariable analysis. Among patients with PAMs, cT3b (OR, 2.667; $p = 0.004$), cT4 (OR, 3; $p = 0.034$) and proportion of BPC (OR, 4.594; $p = 0.027$) were strongly associated with PAMs by using a dichotomous logistic regression univariable model, and cT3b (OR, 3.899; $p = 0.02$), cT4 (OR, 2.8; $p = 0.041$) and proportion of BPC (OR, 5.247; $p = 0.04$) were independently associated with PSMs by multivariable analysis. Conclusions: Patients with nmPCa in our institute had high risk, high ISUP grade and high clinical stage. Tumor biological factors were strongly associated with PSMs and PAMs, and PLN invasion was independently associated with PSMs. The risk factors influenced the status of surgical margins, and apical margins were different.

**Keywords:** radical prostatectomy; positive surgical margins; positive apical margins; pelvic lymph nodes invasion; prostate cancer

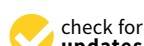



## 1. Introduction

RP is an effective option for treating nmPCa, as it aims to completely remove cancers. Therefore, many patients have PSMs after RP; it is considered an adverse factor associated with prostate specific antigen (PSA) biochemical recurrence (BCR) and poor prognosis [1],

and an independent predictor of disease progression [2]. The rates of PSMs varied differently from 6.5% to 38% in literatures [2,3]; many factors including surgeon's experience and tumor behavioral factors may influence the status of surgical margins [2,4], but the conclusion is still controversial. Apex is the most common PSMs site [5], but which factors associated with PAMs are still unclear.

The objective of this study is to assess the tumor biological features in men with nmPCa in our institute and to determine the association of clinical and tumor biological factors with PSMs and PAMs.

## 2. Material and Methods

The study was retrospective; 200 patients with nmPCa who underwent RP, including 198 with laparoscopic RP and 2 with open RP, at Peking University Cancer Hospital & Institute between January 2013 and December 2017 were reviewed. MRI, Emission computed tomography (ECT) or CT were performed before surgery to confirm no bone or distant organ metastasis. Age (years), TPV (mL), BPC (proportion), D'Amico risk classification, clinical T stage, preoperative basal levels of PSA (ng/mL), ISUP grade, f/t and PLN invasion were assessed and calculated for each case. Extra-fascial radical prostatectomy through an extraperitoneal approach was performed by skilled and experienced surgeons in our institute (at least 200 cases of RP were performed), according to the technique of Walsh et al. [6]; standard PLN dissection with the scope of obturator nerve and internal iliac vessels was performed in all cases, and men with PLN invasion were assessed.

Prostate biopsies should be performed at least 30 days before RP, and the following features should be confirmed in cases of biopsy performed elsewhere: (1) at least 12 biopsy cores; (2) number of positive cores should be reported; (3) Gleason Score should be reported. Ultrasound guided a 13-core trans-rectal prostate biopsy technique that was used in our institute, and the proportion of BPC and Gleason Score were calculated. Tumors were classified into 1–5 grades according to the ISUP 2014 grade group system [7] and classified into the low, intermediate or high grade group according to D'Amico risk classification [8]. Patients enrolled were staged according to the 2010 American Joint Committee on cancer system (AJCC, clinical stage T1–T4) [9]. All specimens were assessed by a pathologist; the PSMs were defined as a tumor extending to the inked surface of the specimen [10]. According to the extension of the tumor invasion (including site of apex, peripheral and base), surgical margins were classified as negative and positive, and for most cases with PSMs that were PAMs, we further analyzed the clinical and pathological data in patients with PAMs, separately.

## 3. Statistical Analysis

The software used to run the analysis was IBM–SPSS version 20. Clinical and pathological data were expressed as frequencies and means. In study groups, differences were assessed by Student's *t* test for continuous variables and by the Chi-squared test for categorical variables. The binary logistic regression model (univariate and multivariate analysis) was used to evaluate the association between significant clinical factors and risk of PSMs or PAMs, which were all compared to negative cases (reference group). All statistical tests were two-sided with a significance level of 0.05.

## 4. Results

### 4.1. Patients' Clinicopathologic Characteristics

A total of 200 patients with nmPCa were enrolled. The median values of clinical and pathological factors were $65.69 \pm 5.96$ years for age, $39.88 \pm 27.96$ mL for TPV, $43.83 \pm 26.21\%$ for proportion of BPC, $30.6 \pm 51.16$ ng/mL for tPSA and $0.14 \pm 0.27$ for f/t. The overall PSMs rate was 53%, which was much higher compared with the 16.6% reported by a systematic review of cases published in 2010 [11], and among men with PSMs, 83% were PAMs. According to D'Amico's risk criteria, risk grades were distributed as follows: 11 (5.6%) were low risk, 10 (5%) were intermediate risk and 179 (89.4%) were

high risk. According to the AJCC system, 4 (2%) were cT1, 84 (42%) were cT2, 58 (29%) were cT3a, 49 (24.5%) were cT3b and 5 (2.5%) were cT4. According to the ISUP 2014 grade group system, 25 (12.5%, GS ≤ 6) were grade 1, 58 (29%, GS 3 + 4) were grade 2, 41 (20.5%, GS 4 + 3) were grade 3, 24 (12%, GS 8) were grade 4 and 52 (26%, GS ≥ 9) were grade 5. PLN invasion was detected in 21 patients (10.5%). Clinical and pathological characteristics are summarized in Table 1. For most of patients with PSMs, they were also PAMs; we further analyzed clinical and pathological data in patients with PSMs and PAMs, separately, to confirm which factors were associated with PSMs or PAMs.

**Table 1.** Distribution of clinical and pathological factors in the population and subgroups of patients according to surgical margins status.

| Variables | Population (*n* = 200) | Surgical Margin | | *p* Value |
| --- | --- | --- | --- | --- |
| | | NSMs (*n* = 94; 47%) | PSMs (*n* = 106; 53%) | |
| Age, years, median (IQR) | 65.69 ± 5.96 | 65.88 ± 6.22 | 65.53 ± 5.75 | 0.693 |
| TPV, mL, median (IQR) | 39.88 ± 27.96 | 44.14 ± 35.28 | 36.2 ± 19.04 | 0.059 |
| BPC (proportion), median (IQR) | 43.83 ± 26.21 | 40.43 ± 25.03 | 46.9 ± 27.04 | 0.153 |
| Class risk, *n* (%) | | | | *p* < 0.001 |
| Low | 11 (5.6) | 10 (10.6) | 1 (0.9) | |
| Intermediate | 10 (5) | 9 (9.6) | 1 (0.9) | |
| High | 179(89.4) | 75 (79.8) | 104 (98.1) | |
| Clinical stage (cT), *n* (%) | | | | *p* < 0.001 |
| cT1 | 4 (2) | 5 (5.3) | 0 (0) | |
| cT2 | 84 (42) | 60 (63.8) | 25 (23.6) | |
| cT3a | 58 (29) | 22 (23.4) | 36 (34) | |
| cT3b | 49 (24.5) | 7 (7.4) | 40 (37.7) | |
| cT4 | 5 (2.5) | 0 (0) | 5 (4.7) | |
| ISUP grade group, *n* (%) | | | | *p* < 0.001 |
| Grade 1 | 25 (12.5) | 21 (22.3) | 6 (5.7) | |
| Grade 2 | 58 (29) | 37 (39.4) | 23 (21.7) | |
| Grade 3 | 41 (20.5) | 20 (21.3) | 19 (17.9) | |
| Grade 4 | 24 (12) | 8 (8.5) | 16 (15.1) | |
| Grade 5 | 52 (26) | 8 (8.5) | 42 (39.6) | |
| PSA level, ng/mL, median (IQR) | 30.6 ± 51.16 | 22.77 ± 56.84 | 37.27 ± 45.02 | 0.062 |
| f/t, median (IQR) | 0.14 ± 0.27 | 0.18 ± 0.4 | 0.1 ± 0.07 | 0.133 |
| PLN invasion, *n* (%) | | | | 0.01 |
| No | 179(89.5) | 92 (97.9) | 90 (84.9) | |
| Yes | 21(10.5) | 2 (2.1) | 16 (15.1) | |

PSMs: positive surgical margins; NSMs: negative surgical margins; BPC: biopsy positive cores; TPV: total prostate volume; PLN: pelvic lymph nodes; ISUP: international of society of urologic pathology.

### 4.2. Positive Surgical Margins vs. Negative Surgical Margins

Initially, clinical and pathology data were analyzed and compared among patients with PSMs and patients with negative surgical margins (NSMs) by Student's *t* test and the Chi-squared test. Among groups, patients with PSMs had a higher D'Amico risk grade (*p* < 0.001), a higher clinical T stage (*p* < 0.001), a higher ISUP grade (*p* < 0.001) and higher rates of PLN invasion (*p* = 0.01), compared with patients with NSMs, while the distribution of age, TPV, BPC and PSA level did not show any significant difference as shown in Table 1.

In univariable analysis, high risk (OR, 1.439; 95% CI, 1.051–1.971), late clinical T stage including cT3a (OR, 1.737; 95% CI, 0.988–3.054), cT3b (OR, 5.286; CI, 2.356–11.856) and cT4 (OR, 6.12; 95% CI, 3.55–12.85), high ISUP grade including Grade 4 (OR, 2; 95% CI, 1.207–4.955) and Grade 5 (OR, 6.167; 95% CI, 2.603–14.611) and PLN invasion (OR, 6; 95% CI, 1.343–26.808) were associated with PSMs, as shown in Table 2.

**Table 2.** Clinical and pathological factors associated with risk of positive surgical margins by the binary logistic regression model.

| Variables | Univariable Analysis Margin Positive vs. Negative | | | Multivariable Analysis Margin Positive vs. Negative | | |
|---|---|---|---|---|---|---|
| | OR | 95% CI | *p* Value | OR | 95% CI | *p* Value |
| Class risk | | | | | | |
| Low | 1(Ref) | 1(Ref) | | 1(Ref) | 1(Ref) | |
| Intermediate | 0.125 | 0.016–0.99 | 0.06 | 0.788 | 0.059–10.611 | 0.857 |
| High | 1.439 | 1.051–1.971 | 0.023 | 6 | 1.343–26.808 | 0.019 |
| Clinical stage | | | | | | |
| cT1 | 1(Ref) | 1(Ref) | | 1(Ref) | 1(Ref) | |
| cT2 | 0.442 | 0.271–0.723 | 0.001 | 1.515 | 0.287–8.004 | 0.625 |
| cT3a | 1.737 | 0.988–3.054 | 0.045 | 5.116 | 1.014–25.802 | 0.048 |
| cT3b | 5.286 | 2.356–11.856 | <0.001 | 9.194 | 1.798–47.017 | 0.008 |
| cT4 | 6.12 | 3.55–12.85 | <0.001 | 4.58 | 2.125–10.45 | 0.01 |
| ISUP grade group | | | | | | |
| Grade 1 | 1(Ref) | 1(Ref) | | 1(Ref) | 1(Ref) | |
| Grade 2 | 0.625 | 0.357–1.093 | 0.099 | 2.388 | 0.457–12.469 | 0.419 |
| Grade 3 | 0.944 | 0.487–1.833 | 0.866 | 3.888 | 0.757–19.976 | 0.104 |
| Grade 4 | 2 | 1.207–4.955 | 0.034 | 7.04 | 1.142–12.379 | 0.035 |
| Grade 5 | 6.167 | 2.603–14.611 | <0.001 | 16.514 | 2.887–29.459 | 0.002 |
| PLN invasion, *n* (%) | | | | | | |
| No | 1(Ref) | 1(Ref) | | 1(Ref) | 1(Ref) | |
| Yes | 6 | 1.343–26.808 | 0.019 | 5.516 | 1.183–25.719 | 0.03 |

Ref: reference group, negative surgical margins; ISUP: International Society of Urologic Pathology; PLN: pelvic lymph nodes.

Independent factors associated with PSMs were estimated using multivariable logistic regression, as shown in Table 2. High risk (OR, 6; 95% CI, 1.343–26.808), high clinical T stage including cT3a (OR, 5.116; 95% CI, 1.014–25.802), cT3b (OR, 9.194; 95% CI, 1.798–47.017) and cT4 (OR, 4.58; 95% 2.125–10.45), high ISUP grade including Grade 4 (OR, 7.04; 95% CI, 1.142–12.379) and Grade 5 (OR, 16.514; 95% CI, 2.887–29.459) and PLN invasion (OR, 5.516; 95% CI, 1.183–25.719) were independently associated with PSMs, as shown in Table 2.

*4.3. Positive Apical Margins vs. Negative Apical Margins*

For 83% of cases with PSMs, they were also PAMs; we analyzed the data among men with PAMs separately to assess the factors associated with PAMs. The overall rate of PAMs was 44% (88 cases). Clinical and pathological data were analyzed and compared among patients with PAMs and patients with negative apical margins (NAMs) by using Student's *t* test and the Chi-squared test. Among groups, patients with PAMs had a higher proportion of BPC (*p* = 0.025), higher frequency of high-risk grade (*p* = 0.005), a later clinical T stage (*p* < 0.001), a higher ISUP grade (*p* = 0.001) and higher rates of PLN invasion (*p* = 0.007), compared with men with NAMs, and the distribution of age, TPV, PSA level and f/t did not show any significant differences, as shown in Table 3.

In univariable analysis, clinical T stage, including cT3b (OR, 2.667; 95% CI, 1.374–5.177) and cT4 (OR, 3; 95% CI, 1.312–18.84) and proportion of BPC (OR, 4.594; 95% CI, 1.188–17.77) were associated with PAMs, as shown in Table 4. Independent factors associated with PAMs were estimated using multivariable logistic regression, as shown in Table 4. Late clinical T stage, including cT3b (OR, 3.899; 95% CI, 1.084–9.399) and cT4 (OR, 2.8; 95% 1.82–19.85), and proportion of BPC (OR, 5.247; 95% CI, 0.998–27.576) were independently associated with PAMs, as shown in Table 4.

**Table 3.** Distribution of clinical and pathological factors in the subgroups of patients according to apical margins status.

| Variables | Apical Margin | | *p* Value |
|---|---|---|---|
| | NAMs (*n* = 112; 56%) | PAMs (*n* = 88, 44%) | |
| Age, years, median (IQR) | 65.4 ± 6.27 | 66.06 ± 5.55 | 0.462 |
| Prostate volume, mL, median (IQR) | 41.96 ± 32.5 | 37.12 ± 20.34 | 0.255 |
| BPC (proportion), median (IQR) | 39.71 ± 24.45 | 50.1 ± 27.74 | 0.025 |
| Class risk, *n* (%) | | | 0.005 |
| Low | 9 (8) | 1 (1) | |
| Intermediate | 9 (8) | 1 (1) | |
| High | 94(84) | 86(98) | |
| Clinical stage (cT), *n* (%) | 112 | 88 | <0.001 |
| cT1 | 4 (3.6) | 0 (0) | |
| cT2 | 63 (56.3) | 21 (23.9) | |
| cT3a | 31 (27.7) | 27 (30.7) | |
| cT3b | 13 (11.6) | 36 (40.9) | |
| cT4 | 1 (0.9) | 4 (4.5) | |
| ISUP grade group, *n* (%) | 112 | 88 | 0.001 |
| Grade 1 | 19 (17) | 6 (6.8) | |
| Grade 2 | 41 (36.6) | 17 (19.3) | |
| Grade 3 | 23 (20.5) | 18 (20.5) | |
| Grade 4 | 11 (9.8) | 13 (14.8) | |
| Grade 5 | 18 (16.1) | 34 (38.6) | |
| PSA level, ng/mL, median (IQR) | 24.36 ± 52.43 | 38.65 ± 48.64 | 0.067 |
| f/t, median (IQR) | 0.16 ± 0.37 | 0.11 ± 0.08 | 0.24 |
| PLN invasion, *n* (%) | 112 | 88 | 0.007 |
| No | 106(96.4) | 73(83) | |
| Yes | 6(5.4) | 15(17) | |

PAMs: positive apical margins; NAMs: negative apical margins; BPC: biopsy positive cores; TPV: total prostate volume; PLN: pelvic lymph nodes; ISUP: International of Society of Urologic Pathology.

**Table 4.** Clinical and pathological factors associated with risk of positive apical margins by the binary logistic regression model.

| Variables | Univariable Analysis Apex Positive vs. Negative | | | Multivariable Analysis Apex Positive vs. Negative | | |
|---|---|---|---|---|---|---|
| | **OR** | **95% CI** | *p* **Value** | **OR** | **95% CI** | *p* **Value** |
| Class risk | | | | | | |
| Low | 1(Ref) | 1(Ref) | | 1(Ref) | 1(Ref) | |
| Intermediate | 0.125 | 0.016–0.999 | 0.05 | 0.247 | 0.026–2.317 | 0.221 |
| High | 0.894 | 0.656–1.218 | 0.478 | 1.2 | 0.605–2.381 | 0.602 |
| Clinical stage | | | | | | |
| cT1 | 1(Ref) | 1(Ref) | | 1(Ref) | 1(Ref) | |
| cT2 | 0.339 | 0.202–0.571 | <0.001 | 0.129 | 0.013–1.325 | 0.085 |
| cT3a | 0.857 | 0.497–1.479 | 0.579 | 0.286 | 0.028–2.93 | 0.292 |
| cT3b | 2.667 | 1.374–5.177 | 0.004 | 3.899 | 1.084–9.399 | 0.02 |
| cT4 | 3 | 1.312–18.84 | 0.034 | 2.8 | 1.82–19.85 | 0.041 |
| BPC | 4.594 | 1.188–17.77 | 0.027 | 5.247 | 0.998–27.576 | 0.04 |
| ISUP grade | | | | | | |
| Grade 1 | 1(Ref) | 1(Ref) | | 1(Ref) | 1(Ref) | |
| Grade 2 | 0.5 | 0.288–0.867 | 0.014 | 0.469 | 0.193–1.142 | 0.096 |
| Grade 3 | 0.45 | 0.205–0.988 | 0.047 | 0.404 | 0.141–1.155 | 0.091 |
| Grade 4 | 1.8 | 0.831–3.899 | 0.136 | 1.5 | 0.534–4.214 | 0.442 |
| Grade 5 | 1.2 | 0.605–2.381 | 0.602 | 1.6 | 0.782–3.38 | 0.782 |
| PLN invasion | | | | | | |
| No | 1(Ref) | 1(Ref) | | 1(Ref) | 1(Ref) | |
| Yes | 2 | 0.684–5.851 | 0.206 | 1.665 | 0.491–5.639 | 0.413 |

Ref: reference group, negative apical margins; ISUP: International Society of Urologic Pathology; PLN: pelvic lymph nodes; BPC: biopsy positive cores.

## 5. Discussion

RP is currently the most commonly used therapeutic option for treating nmPCa; studies confirm that post-operative PSMs are closely related to BCR and tumor progression [3,12]. Presence of PSMs usually means further treatments including adjuvant radiotherapy (ART) and/or adjuvant antiandrogen therapy. The most common site of PSMs is the apex [13]. Many factors may influence the status of PSMs; these factors are categorized into two groups: surgeon's experience and tumor behavioral factors [2,4]. Tumor behavioral factors may be more strongly associated with PSMs compared with surgeon's experience. A previous study reported that BMI, PSA level and high D'Amico risk were all independent risk factors associated with PSMs [2,4]. A study reviewed 45,426 patients with T2 stage for whom RP was performed in 1152 institutes, and found that Gleason Score, tPSA level and nations were positively correlated with PSMs, while saturation of surgery was negatively correlated with PSMs [14]. Another study suggested that total testosterone (TT) level was important for predicting PSMs; by using a multinomial logistic regression model, the study confirmed that TT was associated with PSMs, and by using multivariable analysis, it confirmed that TT was the only independent factor associated with PSMs [15]. Recently, a study suggested that level of invasion into the fibromuscular band of prostate was strongly associated with PSMs [16]; in this study, each specimen was examined in 3–5 mm sections from base to apex, perpendicular to the major; then, slide-mounted thin sections were stained, and the percentage of the tumor volume was categorized into three groups: <5%, 5–15% and >15%; the results demonstrated that the level of invasion into the fibromuscular band was an independent risk factor for PSMs. Therefore, tumor biological factors strongly influenced the rate of PSMs, but which factor mostly associated with PSMs remained controversial. Surgeon's experience was also thought to be associated with PSMs in some literatures; leaning curves for surgical margins after open or laparoscopic RP plateaued at approximately 200–250 cases, as reported [17–19]. Some other studies suggested that there was no association of surgeon's experience with PSMs [20]. Therefore,

there was no clear conclusion on the risk factors that influenced the status of PSMs; further exploration was needed.

The reported rates of PSMs after RP varied dramatically from 6.5% to 38% in contemporary studies [2,3]. Rate of PSMs in patients who underwent open RP varied from 11% to 38%, while in patients who underwent da Vinci laparoscopic assistant RP varied from 12% to 32.8% [4,21]; the type of surgery seemed not to be the decisive factor for PSMs. Rate of PSMs in our study was 53% which was much higher than reported, although all RP according to the technique of Walsh were performed by experienced surgeons (at least 200 cases of RP were performed), and we were trying to find the factors led to the results. The distribution of clinical and pathologic factors in our study had their own features; 56% were pT3-4, 89.4% were high risk, 58.5% were ISUP 3–5 grade, and that might lead to the high rate of PSMs in our study, and the features were consistent with the characteristics of localized PCa in China, as reported by a study that indicated that 77.4% of patients in China had intermediate- or high-risk disease according to the Cancer of the prostate risk assessment post-surgical score, and 87.5% had a Gleason score $\geq$8 [22], while a recent national database from the United States showed high risk attribution in only 6.5% of cases [23]. According to univariable analysis, our results concluded that high risk, cT3-4, ISUP grade 4–5 and PLN invasion were associated with PSMs. Furthermore, high risk, cT3-4, ISUP grade 4–5, andPLN invasion were all independent risk factors associated with PSMs. Therefore, tumor biology was strongly associated with PSMs, which might lead to a high rate of PSMs in our study.

The prostatic apex represented the most common PSM site after RP, as reported by many literatures [5,24]. Apex dissection was an important and challenging step of the RP procedure; surgeons needed to dissect as much prostate tissue as possible, while reserving the sphincter to avoid urinary incontinence after surgery, and in patients with PAMs treated with ART after RP, latent urinary incontinence would occur [25], so detection of the risk factor associated with PAMs was important. Different explanations were formulated to explain why positive margins were more common at the apex, but the conclusion was still controversial. Findings from our analysis confirmed that the apex was a frequency location of PSMs in patients with PSMs 83% that were PAMs. According to univariable analysis, risk factors including cT3b, cT4 and proportion of BPC were associated with PAMs, while cT3b, cT4 and proportion of BPC were independent risk factors associated with PAMs, according to multivariable analysis.

## 6. Conclusions

Tumor biological features of our study were consistent with the characteristics of localized PCa in China; as reported, patients had a high Gleason Score, high ISUP grade, high risk grade and late clinical T stage. Tumor biology was strongly associated with PSMs and PAMs, which was consistent with literatures as reported [26–28], and in our study we found that PLN invasion (involvement of lymph nodes around the obturator nerve and internal iliac vessel) was an independent risk factor strongly associated with PSMs, and no relative researches was reported in recent years. The factors that influenced the status of surgical margins and apical margins were different; clinical T stage, D'Amico risk grade, ISUP grade and PLN invasion were powerful predictors of PSMs, while clinical T stage and proportion of BPC were powerful predictors of PAMs. Our study had important preoperative implications for surgeons to evaluate the status of surgical margins and apical margins, and patients who showed a high risk of PSMs or PAMs should be informed about the option of undergoing further treatment after RP.

**Author Contributions:** P.D. and S.W. designed the study. S.W., Y.C., X.Y. and Y.Y. performed the study and analyzed the data. S.W. and P.D. wrote the manuscript draft and revised the manuscript. All authors have read and agreed to the published version of the manuscript.

**Funding:** This research received no external funding.

**Institutional Review Board Statement:** The study was conducted according to the guidelines of the Declaration of Helsinki, and approved by the Institutional Review Board of Peking University Cancer Hospital & Institute in April 2020 (protocol code 2018KT27).

**Informed Consent Statement:** Patient consent was waived due to the retrospective nature of the study.

**Data Availability Statement:** The data presented in this study are available on request from the corresponding author. The data are not publicly available due to privacy of patients.

**Conflicts of Interest:** The authors declare no competing interests.

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
