# Peer review of "Tumor Biological Feature and Its Association with Positive Surgical Margins and Apical Margins after Radical Prostatectomy in Non-Metastasis Prostate Cancer"

_curroncol, doi:10.3390/curroncol28020144_

Round 1

Reviewer 1 Report

An interesting publication, mainly because of the differences concerning the Chinese population. The authors evaluated patients with prostate cancer, most of whom had high or very high risk cancer. Comments: - The study is retrospective and covers the period 2013-2017 (5 years). During this time, 200 patients were operated on, which amounts to 40 patients per year. This is not the number of operations characterising a high-volume centre - surgeries were performed from a pre-peritoneal access, which, given the stage of the disease in most patients, makes it impossible to perform a proper pelvic lymphadenectomy. This may account for the low number of patients with lymph node involvement (only 10.5%). This number would probably be higher if surgery was performed transperitoneally. -taking the above into account the determination of the relationship between the selected characteristics is controversial In my opinion, this publication, although interesting, is based on inconclusive data. It would require either a separate explanation from the authors (a letter to the editor?) or rejection.

Author Response

Dear Professor and Editor

   Really thanks for your reply and suggestions! Following your comments, I have modified my paper, please take your time to check, thank you!

  • Moderate English changes required.

Re: Following your suggestions I have modified my English writing in the paper, all the modified place are by red words.

  • Does the introduction provide sufficient background and include all relevant references? – can be improved

Abstract has been modified with red words,

line 42 “PSMs is found in some patients after RP” changed to “many patients have PSMs after RP.”

Line 44 (deleted “it is”); line 45 (added “6/5% to 38% in”); ling 46-47 (2 references were added and “including surgeon’s experience and tumor behavioral factors” was added).

  • Is the study design appropriate? – must be improved

Thank you, it is really a good suggestion! Your suggestion may be based on the insufficient scope of pelvic lymph nodes dissection may lead to the inconclusive data.

Due to guideline for PCa improved very quickly, the suggested scope and purpose for pelvic lymph nodes dissection (PLND) in PCa were changed in recently years. The purpose for PLND used to determine the tumor stage rather than treatment effect, the standard dissection is recommended rather than extended dissection until recent years, in newest guideline (2020) extended PLND is recommended for advanced and high risk nmPCa. In our institute standard PLND was used for all non-metastatic PCa enrolled in this study (2013-2017) following guideline, indeed compared with extended PLND standard PLND may lead to the false negative result, but majority of prostate tumor firstly invaded lymph nodes around obturator nerve and internal iliac vessels (standard PLND), then the common iliac vessels (extended PLND), so if the lymph nodes around internal iliac vessels and obturator nerve are not invaded, the lymph nodes around common iliac vessels are less likely to be involved, therefore, the data in the paper can objectively reflect the situation of pelvic lymph nodes involvement. Although most nmPCa are high risk, the rate of lymph node involvement is low, which may be a characteristic of Chinese patients. In addition, more than 50% of PCa in China especially in Beijing have distant metastases when they first defined as PCa, and surgery is not recommended for mPCa, that’s why there are so few PCa surgeries in our institute. Thanks again for your suggestion! Please contact me if you have any further questions!

  • Are the methods adequately described? - can be improved

Methods have been modified following your comments.

Line 56-57 (added “MRI, Emission computed tomography(ECT) or CT were performed before surgery to confirm no bone or distant organ metastasis”),

Line 60 (added “assessed and”),

line 63-64(added “with the scope of obturator nerve and internal iliac vessels”).

  • Are the results clearly presented? – can be improved

Results have been modified following your suggestions.

Line 91 (in→by)ï¼›Line 102(and→or)ï¼›line 112 (added “while”)ï¼›line 114( high→late)ï¼›Line 122 (,→and)ï¼›Line 130 (surgical→apical)ï¼›line 134(added “using”)ï¼›Line 136(higher→later)ï¼›Line 147(high→late)ï¼›in Table1, p=0.001 changed to 0.001. Thanks!

  • Are the conclusions supported by the results? – can be improved

Conclusions have been modified following your suggestions.

Line 215 (high→late)ï¼›

Line 271 (added “involvement of lymph nodes around obturator nerve and internal iliac vessel).

   I have modified the paper following the suggestions you given in this paper, thanks again for your suggestions and comments!!

Reviewer 2 Report

It is unclear by which procedure they were diagnosed "... men with non-metastasis prostate cancer (nmPCa) ..."

I just believe that the work is good, yes, but that in the definition of "patients without metastases", the diagnostic method used to define the patient without metastasis must be specified (PET, MRI, bone scan?). This calls into question the whole selection of patients and therefore the research setting.

Author Response

Dear Professor and Editor

  • Does the introduction provide sufficient background and include all relevant references? – must be improved

Abstract has been modified with red words,

line 42 “PSMs is found in some patients after RP” changed to “many patients have PSMs after RP.”

Line 44 (deleted “it is”); line 45 (added “6/5% to 38% in”); ling 46-47 (2 references were added and “including surgeon’s experience and tumor behavioral factors” was added).

  • Is the study design appropriate? – Yes

Appreciate for your approval.

  • Are the methods adequately described? - must be improved

Thanks for your comments. In this part your major concern is I did not mention the diagnostic method used for determine the “non-metastatic prostate cancer”, that is my mistake for forgetting to mention that. In line 56-57, the paper was modified to mention that “MRI, Emission computed tomography(ECT) or CT were performed before surgery to confirm no bone or distant organ metastasis”.

Line 60 (added “assessed and”),

line 63-64(added “with the scope of obturator nerve and internal iliac vessels”).

  • Are the results clearly presented? – Yes

Appreciate for your approval.

  • Are the conclusions supported by the results? – Yes

Appreciate for your approval.

   I have modified the paper following the suggestions you given in this paper, thanks again for your suggestions and comments!!

Round 2

Reviewer 1 Report

The authors have improved their article in a significant way.
There are still some uncertainties regarding the type of lymphadenectomy performed and the percentage of lymph nodes involved, especially in patients with high-risk cancer. Nevertheless, the article as a whole is suitable for publication in its present form. The article still requires English language correction.

Reviewer 2 Report

The absence of metastases has been shown properly